# Combined Exposure to Diazinon and Nicotine Exerts a Synergistic Adverse Effect In Vitro and Disrupts Brain Development and Behaviors In Vivo

**DOI:** 10.3390/ijms22147742

**Published:** 2021-07-20

**Authors:** Bonn Lee, Seon Mi Park, SunHwa Jeong, KangMin Kim, Eui-Bae Jeung

**Affiliations:** Laboratory of Veterinary Biochemistry and Molecular Biology, College of Veterinary Medicine, Chungbuk National University, Cheongju 28644, Korea; bonnleedvm@gmail.com (B.L.); qkrtjsal0321@naver.com (S.M.P.); tnsghk0202@naver.com (S.J.); jfun4my@gmail.com (K.K.)

**Keywords:** organophosphate, combined exposure, diazinon, nicotine, dopaminergic neuron

## Abstract

A real-life environment during pregnancy involves multiple and simultaneous exposures to toxic chemicals. Perinatal exposures to toxic chemicals have been reported to exert an inhibitory effect on mouse neural development and behaviors. However, the effect of combined exposures of organophosphate and nicotine has not been previously reported. In this study, we investigated whether a combined exposure of diazinon and nicotine can have a synergistic effect. The effects of the combined chemical exposure on cell viability and neuronal differentiation were examined using mouse Sox1-GFP cells. Additionally, mice were maternally administered 0.18 mg/kg diazinon, a no adverse effect level (NOAEL) dose, combined with 0.4, 1, and 2 mg/kg nicotine. Mice offspring underwent behavior tests to assess locomotor, depressive, cognitive, and social behaviors. Morphological change in the brain was investigated with immunolocalization. We revealed that the combined exposure to diazinon and nicotine can have a synergistic adverse effect in vitro. In addition, the chemical-treated mouse offspring showed abnormalities in motor learning, compulsive-like behaviors, spatial learning, and social interaction patterns. Moreover, 0.18 mg/kg diazinon and 2 mg/kg nicotine co-exposure resulted in an increase in tyrosine hydroxylase (TH)-positive dopaminergic neurons. Thus, the findings suggest that perinatal co-exposure to nicotine and diazinon can result in abnormal neurodevelopment and behavior, even at low-level administration.

## 1. Introduction

During pregnancy, maternal exposure to a specific chemical compound may result in abnormal brain development and behavior in humans and animals, and the real-life environment during pregnancy can involve multiple and simultaneous exposures to toxic chemicals, such as pesticides and cigarette smoke. In this regard, synergistic adverse effects generated by co-exposure of toxicants have attracted contemporary researchers. For example, even at an environmental-level exposure, a mixture of pesticides (varying from 0.001 to 0.004 ppm) can generate an inhibitory effect on aquatic ecosystems via oxidative stress or cholinergic inhibition [1]. Likewise, a combination of benzopyrene and lead (Pb) can produce a synergistic adverse effect on spatial learning and memory impairments by exacerbating oxidative stress [2]. In addition, co-exposure of dichlorvos and monocrotophos has resulted in more severe damage to neurons than from individual-exposures by depleting neurotransmitter levels [3]. Indeed, combined exposure to multiple substances may enhance or counterbalance the toxicity of chemicals, leading to unknown risks to animal health.

Diazinon is one of the most extensively used organophosphate pesticides for household or agriculture purposes [4]. Additionally, cigarettes represent one of the globally abused toxicant sources and cigarette smoke contains various toxic chemicals. There is substantial evidence that nicotine mainly exerts its neurotoxic effect by activating or desensitizing neuronal receptors [5]. Both chemicals have been reported to produce neurotoxicity individually. However, the effect of combined exposures of nicotine and organophosphate has not been previously reported. Therefore, in this study, we investigated an effect of multi-combined exposure to nicotine and diazinon on brain development in vitro and in vivo.

Synergistic adverse effect of multi-chemical exposure to nicotine and diazinon was investigated in Sox1-GFP embryonic stem cells. Cell viability after the chemical exposure was measured, and evaluation of neural differentiation was carried out with the endogenous cell specific reporter (Sox1-GFP), which is a well-known marker for neural stem and progenitor cells [6].

Maternal exposure to nicotine affects both the mental and physical health of offspring, including effects on cognition, locomotion, and learning ability [7]. Therefore, behavior testing is important when examining the effects of chemical exposure on offspring health. In addition, there is evidence indicating that prenatal nicotine exposure can result in remarkable morphological changes in the brain, such as an abnormal brain compartment or the precocious development of neuronal receptors. Herein, we conducted mouse behavior tests and undertook tissue immunostaining to obtain a comprehensive evaluation of mouse health following diazinon and nicotine exposure. Diazinon and nicotine were perinatally injected into maternal mice. No adverse effect level (NOAEL) of diazinon was used, and nicotine was injected at a short-term exposure (four days) during pregnancy. Motor, depression, recognition, and social interaction behaviors were assessed in mice offspring, as well as the evaluation of the morphology of dopamine neurons in the brain.

## 2. Results

### 2.1. Combined Administration of Nicotine and Diazinon Exerted a Synergistic Adverse Effect on Cell Viability of Sox1-GFP Cells

To investigate whether the combined-exposure to nicotine and diazinon has a synergistic effect in neural development, undifferentiated Sox1-GFP cells, mice neural stem cell, were treated with both chemicals simultaneously. After 24 h from the chemical exposure, cell viability was evaluated by an IC_50_ value obtained from the CCK-8 assay (Figure 1A,C,E). The chemical-containing medium was prepared at six different concentrations, from 10^−5^ M to 1 M of nicotine combined with 10^−5^ M of diazinon (no observed adverse effect level; NOAEL), respectively (Figure 1E,F). The IC_50_ value of the combined exposure to nicotine and diazinon indicated a more decreased cell viability (*p* = 0.0283 vs. nicotine only, *p* = 0.0142 vs. diazinon only; Figure 1G) than both the nicotine only-treated and diazinon only-treated groups. Additionally, the protein expression level of Sox1, the neural differentiation marker, was measured by GFP intensity of the stem cells (ID_50_ values). There were no significant differences between the combined chemical-treated group and nicotine only-treated group (Figure 1B,D,F). Our result suggests that the combined exposure can exert a synergistic adverse effect in terms of cell viability but did not inhibit neural differentiation in vitro.

### 2.2. Combined Administration of Nicotine and Diazinon Resulted in Abnormal Locomotor Function and Motor Learning Impairment

To evaluate the effect of combined exposure to nicotine and diazinon on murine brain function, mice offspring underwent behavioral testing. Rotarod and open field tests were performed to evaluate the locomotor function of the mice. All male and female mice were able to train on the rotarod, and there were no remarkable differences among treated groups. However, after the training phase (from day 6 to day 10), both male and female mice in all chemical exposure groups exhibited a marked reduction in latency to fail compared to that of the vehicle-treated group (Figure 2A). NC2 + DZ injected group exhibited significantly reduced motor learning ability in both males and females consistently. The open-field test data was summarized in Figure 2D–G. The co-exposure to nicotine and diazinon exerted an effect only in male mice, while females did not exhibit any modification in open field parameters. NC0.4 + DZ treated male mice frequently accessed the center of the open field cage and spent significantly extended time in the center zone. Together, the results indicate that co-exposure to nicotine and diazinon can partially interrupt locomotor function.

### 2.3. Combined Administration of Nicotine and Diazinon Partially Increased Compulsive-like Behavior in Mice, but Did Not Affect Depression-Related Behaviors

Marble-burying and tail suspension tests were performed to assess the depression and anxiety-like behaviors of mice. The marble-burying test result revealed that mice exposed to 1 mg/kg or 2 mg/kg of nicotine and 0.18 mg/kg diazinon had increased levels of compulsive-like behaviors compared to that of the vehicle-treated control group. Figure 3A shows representative cages from each treatment group and Figure 3B depicts the number of buried marbles. Male mice with 1 mg/kg or 2 mg/kg nicotine exposure buried a significantly increased number of marbles than did the vehicle-treated group (Figure 3B, *p* < 0.05 respectively). Female mice with 2 mg/kg nicotine treated mice also buried a greater number of marbles than VE did. However, contrary to male mice in the NC1 + DZ group, females did not show any significant difference in the test. In addition, nest-building and tail suspension tests were performed to investigate whether combination chemical exposure affected the expression of depression-related behaviors in mice. The test results indicated no significant differences in nest-building ability between all the experimental groups and the vehicle-treated group (Figure 3C). Additionally, immobility time during tail suspension was measured as an index of depressive behavior. However, mice in the experimental groups did not show any significant difference in immobility time from that of the control group (Figure 3D). These results suggest that 2 mg/kg of nicotine and 0.18 mg/kg of diazinon co-exposure can result in compulsive-like behaviors in the offspring.

### 2.4. Combined Exposure of Nicotine and Diazinon Disrupted Spatial Learning and Cognition-Related Behaviors in Mouse Offfspring

Memory and cognition abilities were evaluated using mouse behavior tests. The Morris water maze test was performed to assess spatial learning and memory disturbance. Mice were trained to recognize a visual cue as evidence of the location of a hidden platform. Each mouse underwent four test trials per day over a series of days. The first four consecutive days were considered the acquisition phase. All mice were able to learn the platform location. However, there was a significant difference in escape latency between the vehicle-treated and chemical-treated groups. Both male and female offspring exhibited abnormal spatial learning behavior during the Morris water maze. The delay appeared on the initial phase of Morris-water maze training in both male and female. In male group, mice administrated with 0.4 mg/kg or 2 mg/kg of nicotine combined with diazinon (0.18 mg/kg) had relatively greater latent escape time during the initial phase of learning compared to mice from the control group. On day 3, the 0.4 mg/kg and 2 mg/kg nicotine treatment resulted in a delay of escape times (*p* < 0.05, respectively; Figure 4A). In the probe test (day 10), the platform was removed, and the behaviors of the mice were assessed. The time spent in the area where the platform was previously located, frequency to cross the area, swimming distance, and swimming speed were measured to investigate the spatial recognition capacity of the mice (Figure 4D–G). Among these assessments, the male offspring exposed to the 2 mg/kg nicotine group had a longer swimming distance than that of the vehicle control group (Figure 4D; *p* < 0.05), while other groups did not show any differences compared to vehicle controls. Visual representation images displayed that 2 mg/kg nicotine-injected mice moved farther during the probe test than did the vehicle-treated mice. Considering the observations from the acquisition phase and during the probe test, the results suggest that perinatal exposure to nicotine and diazinon can impair the spatial memory of the mouse offspring. In addition, a novel object recognition test was used to assess whether simultaneous exposure to nicotine and diazinon could influence the recognition memory of mice. A representative schematic of the tests are presented in Figure 4H. Normal rodents tend to spend more time exploring a novel object during the test phase [8]. Contrary to the normal behavior, the 0.4 mg/kg and 2 mg/kg nicotine exposure groups spent more time close to the familiar object (Figure 4I,J), suggesting that combined nicotine and diazinon administration may impair the cognitive ability of offspring mice.

### 2.5. Simultaneous Exposure to Nicotine and Diazinon Disturbed Sociability-Related Behaviors, but Seldom Affected on the Animals’ Social Novelty Recognition

We conducted social interaction and three-chamber sociability tests to evaluate the sociability-related behavior of the mice (Figure 5). A social interaction test was performed. In the test, the amount of general sniffing, anogenital sniffing, mouthing, and following events were determined (Figure 5A–D). We observed a significant reduction of social interaction patterns, such as following and anogenital behavior (Figure 5A,B). Statistical significance was not detected between the two sexes. On the other hand, the three-chamber sociability results revealed that nicotine and diazinon co-treatment do not influence social novelty recognition (Figure 5F–I). Generally, a mouse tends to spend more time closer to a stranger mouse than a familiar object [9]. In the test, all mice groups revealed a strong preference for stranger mice (Stranger1) than the empty cup. Therefore, all the preference indexes depicted positive value (Figure 5F,G). However, in the social novelty test, preference towards stranger 1 or stranger 2 was inconsistent among the mice groups, resulting in a fluctuation in the indexes. The 0.4 mg/kg and 1 mg/kg nicotine treated male mice indicated negative values, while females indicated positive values in all groups. Nevertheless, there was no significant difference among the mice (Figure 5D–E). Conclusively, our findings suggest that nicotine and diazinon co-exposure disturbed sociability-related behaviors but did not affect social novelty recognition.

### 2.6. Combined Exposure of Nicotine and Diazinon Increased the Density of the Dopaminergic Neuron in Ventral Tegmental Area in Male Mice

To examine the effect of a multi-chemical exposure of nicotine and diazinon on brain formation, brain tissues from male offspring mice were analyzed (*n* = 4 for male, *n* = 3 for female). We performed immunostaining to determine if the treatments resulted in morphological changes in the mouse brain. According to previous reports, dopaminergic neurons populate mainly on the ventral tegmental area (VTA) and the substantia nigra pars compacta (SNc) (Figure 6A) [10,11]. The tyrosine hydroxylase (TH) antibody was used to localize dopaminergic neurons. The number of TH-positive cell bodies around the VTA, SNc, and substantia nigra pars reticulata (SNr) was counted with NIH Image J software (Figure 6B). The chemical treatment caused an alteration in the number of cells only in the male group, while female mice exhibited no statistical significance. The number of TH-positive cells in the assessed area was greater in the 2 mg/kg nicotine-treated group than in the other treated groups, indicating that a high dose (2 mg/kg) of nicotine combined with diazinon can modify the number of dopaminergic cells in the VTA, SNc, and SNr brain regions (Figure 6C, *p* < 0.05).

## 3. Discussion

To the best of our knowledge, the combined neurotoxic effects of nicotine and organophosphate have not been reported yet. Therefore, this study aims to examine the effects of multi-chemical exposure on murine neural development in vitro and in vivo. Our observation revealed that simultaneous treatment of nicotine and diazinon exerted a synergistic inhibitory effect on neuronal stem cell viability. In addition, when the chemicals were maternally administered to mice, it resulted in impaired locomotor learning, novel recognition, spatial learning, compulsive-related behavior, and social interaction patterns in offspring mouse. Moreover, we also observed that the amounts of TH+ dopaminergic cells in the VTA, SNc, and SNr region were higher in male offspring in the multi-chemical-treated group. Maternal exposure to toxic chemicals perturbs neonatal development and has lifelong neurobehavior consequences in offsprings [12]. Nicotine and organophosphate are well-known neuroteratogens and single-exposure effects of each of those chemicals have been previously reported. Unlike previous studies, this study highlighted the adverse effect of co-exposure to nicotine and organophosphate.

First, the synergistic adverse effect of the combined exposure was demonstrated in Sox1-GFP cells. Sox1-GFP system was concieved to screen the adverse effect on neural differentiation in mammalian cells [13]. Since the co-exposure to nicotine and diazinon exacerbated the cell viability more severely than a single-chemical-exposure, we continued in vivo study to evaluate the seriousness of the co-exposure to nicotine and diazinon.

Developmental nicotine exposure has resulted in adverse outcomes in vitro and in vivo [14,15,16,17]. Even though test protocols and drug concentrations vary from those in our study, the previous studies support our major observations: (1) reduced motor learning, (2) compulsive behavior, (3) decreased cognition, and (4) alternation in social interaction patterns. Lee and Picciotto (2019) reported that nicotine exposure (delivered thorough drinking water; 200 μg/mL) during development impairs learning behaviors and motor tasks via the modification of cortical synaptic plasticity. Zhou (2021) revealed that maternal nicotine exposure (via drinking water; 200 μg/mL) resulted in increased moving distance in an open field test for offspring mice [17]. Moreover, Alkam (2013) reported that maternal nicotine exposure (via drinking water; 200 μg/mL, *ad libitum*) weakened the emotional behaviors of offspring mice, including those displayed in the marble-burying test. In addition, Li (2015) observed that long-term exposure to nicotine during pregnancy (injected via osmotic implant; 6 mg/kg/day) resulted in a delayed escape latency in the Morris water maze test, indicating spatial learning and cognition abnormalities in the offspring. The alternation of mice social interaction patterns, in perinatal nicotine or diazinon exposure models, are our novel observations.

Furthermore, although we analyzed the data by splitting males and females, we could not demonstrate statistical difference by the sex variable. However, previous reports have shown a correlation between offspring sex and impairment in offspring [15,18,19,20]. Perinatal nicotine exposure has been reported to disrupt motor learning and increase compulsive behavior in male offspring in mice, but not in females. Alkam (2013) reported that perinatal nicotine injection was more vulnerable to male offspring than female in mice, resulting in motor learning deficiency only in male offspring but not in female offspring [15]. It has been documented that perinatal exposure to nicotine interferes with murine CNS development via gonadal hormone-mediated sexual differentiation of the brain during pregnancy [18]. Likewise, prenatal exposure to endocrine-disrupting chemicals (EDCs) preturbs brain development by disrupting sex-specific developmental processes [19]. At last, prenatal diazinon exposure altered the passive avoidance performance and neuronal nitric oxide (nNOS) synthase neurons in basolateral complex of amygdala in female rat, while the males were not affected [20]. In conclusion, previous studies suggest that perinatal exposure to nicotine or diazinon may exert effects in a sex-dependent manner.

On the other hand, perinatal nicotine-induced abnormalities in development and behavior in mice have been associated with dopaminergic neurons that are responsible for cognition (via mesolimbic pathway) and motor and learning abilities (via nigrostriatal pathway) [11,21,22]. Approximately 75% of dopaminergic neurons reside in the ventral midbrain area, and the absolute number of such cells is between 20,000 and 30,000 in adult mice [23]. Herein, we assessed the number of TH-positive cells in the VTA, SNc, and SNr brain regions. We found that the number of TH-positive dopaminergic neurons increased in the 2 mg/kg nicotine and 0.18 mg/kg diazione-treated male mice. Interestingly, our observation is identical to that in a previous report by Lai et al. (2020): chronic injection of nicotine (50 mg/L, delivered by drinking water) increased the number of TH+ cells in the SNc region [24]. However, the exact mechanism by which nicotine promotes a dopaminergic neuron increase remains unclear. Lai (2020) suggested that the increased TH+ cells were recruited from an endogenous reserve pool of neurons in the SNr, insisting that the increase in the TH+ neurons was not due to cell proliferation or migration. Conversely, Justin (2018) reported that nicotine itself could have a protective effect and promote neuronal survival by suppressing SIRT6, an enzyme related to neurodegeneration [25]. Additionally, maternal nicotine exposure (1.5 mg/kg/day) can stimulate the neurogenesis of orexin-producing neurons in the hypothalamus in offspring [26]. Nicotine exposure at the developmental stage can enhance or inhibit neurogenesis in a region-specific manner [27]. Furthemore, the number of neurons in a specific region is important. Perinatal nicotine exposure generates changes in the cell number of various neuronal cells, resulting in unusual behaviors in the affected animal [28]. Admittedly, though there is a lack of direct evidence to connect our morphological finding with regard to the abnormal behaviors observed in this study, there is indirect evidence that developmental nicotine exposure stimulates a morphological shift of neurons by regulating histone methyltransferase activity via the *Ash21* gene [29]. In addition, it has been reported that maternal nicotine exposure sex-dependently altered offspring striatal dopaminergic system development and resulted in a decrease in 3.4-dihydroxyphenylacetic acid (DOPAC), the metabolite of dopamine, in the midbrain in male mice [30,31].

Diazinon is a widely employed organophosphate insecticide and, in general, organophosphates produce adverse effects on health by disturbing cholinesterase dynamics or by directly producing neurotoxicity via oxidative stress or neural inflammation [32]. In other studies, it has been reported that maternal exposure to diazinon caused abnormal behaviors in offspring. Chronic perinatal exposure to 1 mg/kg or 2 mg/kg diazinon has disturbed the cognitive behavior of offspring, but it did not influence other behaviors such as the fear-response or spatial learning memory [33]. Interestingly, despite the fact that the dose of diazinon in this study was below 0.2 mg/kg (NOAEL: [34]), the mice exhibited abnormal behaviors.

The concentrations of each drug employed in the present study were selected to reflect a low level that could mimic natural nicotine or pesticide exposure in the environment. Nevertheless, our results indicate that their combined administration can result in abnormalities in both behavior and brain morphology. In many studies, combined exposure to multiple substances can inflate the toxicities of the chemicals, leading to brain damage. For example, a combination of benzopyrene and lead caused spatial learning and memory impairments by amplifying oxidative stress levels [2]. Furthermore, by depleting neurotransmitter levels, co-exposure of dichlorvos and monocrotophos resulted in more severe damage to neurons than from individual exposure [3]. The present study indicates that there is synergism in the action of nicotine and diazinon in vitro, and perinatal exposure to both nicotine and diazinon resulted in abnormal mice behaviors and alteration in TH-positive neurons in the brain. In addition to previous reports on the effect of nicotine or diazinon on mouse behaviors, the present findings provide a broader understanding of their effects. However, there is a limitation in that nicotine or diazinon single-exposure data were not included in the study. Accordingly, the seriousness of exposure to respective chemicals and the mechanism through which they act was not elucidated in the present study. Therefore, further studies should focus on how the two chemicals interact.

## 4. Materials and Methods

### 4.1. Mouse Embryonic Stem Cell Culture

The Sox1-GFP mouse embryonic stem cells were kindly donated by Dr. Eekhoon Jho (Laboratory of Cellular Signaling Transduction, University of Seoul, Republic of Korea). The cells were cultured in a 100-mm plate (Corning, NY, USA) coated with 0.2% gelatin in a humidified culture incubator (37 °C with 5% CO_2_). The growth medium consisted of DMDM (Gibco, Grand Island, NY, USA), penicillin (100 U/mL)/streptomycin (100 mg/mL; Cat#: L0022-001; Biowest, Rue de la Caille, Naaille, France), 15% fetal bovine serum (FBS; Biowest), MEM Non-Essential Amino Acids (NEAA, Gibco), and mouse leukemia inhibitory factor (mLIF; Cat#: ESG1106; Milipore, Burlington, MA, USA). To maintain the undifferentiated cell, mLIF was added to the growth medium. For the neuronal differentiation of Sox1-GFP cells, DMDM/F12 (Gibco) supplemented with L-glutamine (Cat#: LS25030081; Gibco), 2-mercaptoethanol (Cat#: 21985-023; Gibco), penicillin/streptomycin, N2 supplement (Cat#: 17502048; Gibco), B27 (Cat#: 17504044; Gibco), and bovine serum albumin (BSA) fraction V (7.5%) (Cat#: 15260-037; Gibco) were cultured in 96-well round bottom plates (Corning).

### 4.2. Cell Viability Assay

Cells were cultured in growth medium. Briefly, 70 μL of cell suspension containing 0.7 × 10^4^ cells per well were plated in 96-well plates. After 24 h, the medium was discarded and replaced with 200 μL growth medium, with/without the nicotine or diazinon, and cultured for 48 h. After the treatment period, cells were rinsed with DPBS and incubated with cell counting kit-8 (CCK-8; Cat#: CK04; Dojindo, Muinch, Germany) solution at 37 °C for an hour. The absorbance of each well was measured at 450 nm, with an Epoch Microplate Spectrophotometer (BioTek, Winnoski, VT, USA). The viability of the cells was normalized by that of control cells, which was fitted as 100%. The half maximal inhibitory concentration (IC_50_) values were determined according to the concentration-response curve using GraphPad Prism (v.5.0; GraphPad Software, San Diego, CA, USA).

### 4.3. Assessment for Neural Differentiation in Sox1-GFP Cells

Cells were cultured in differentiation medium. Specifically, 140 mL of cell suspension containing 100 cells per well with/without nicotine and diazinon were thawed in 96-well round bottom plates (Corning). After culturing for 4 days, a neurosphere was formed in each well. During differentiation, the neurospheres expressed the neuronal differentiation marker, Sox1 protein, and the protein expression level of Sox1 was evaluated by the green fluorescent protein (GFP) intensity. The intensity of GFP in neurospheres was measured with the Lionheart FX Automated Microscope (BioTek) and normalized by the area of each neurosphere. The half inhibition concentration for neural differentiation (ID_50_) values were calculated using GraphPad Prism (v.5.0, GraphPad Software).

### 4.4. Animals and Drug Treatments

Specific pathogen-free C57BL/6J mice (8 weeks, male and female, 25–30 g) were obtained from Koatech (Pyeongteak, Republic of Korea). All animals were nurtured in polycarbonate cages and acclimated to an environmentally controlled room. Detailed information regarding the maintenance was as previously described [35]. After the acclimatization, female mice were mated with male mice overnight at a proportion of 2:1, and the presence of a vaginal plug was set as embryonic day (E) 0.5. The maternal mice were randomly divided into four groups, including the vehicle control group and three exposure groups (*n* = 5 mouse per group). From the embryonic day 9.5 to 12.5, the maternal mice were administrated orally with (-)-nicotine (NC; 0.4, 1, 2 mg/kg; Cat#: N3876, Sigma-Aldrich, St. Louis, MO, USA) and simultaneously with diazinon (DZN; 0.18 mg/kg; Cat#: 492175, Sigma-Aldrich) for four consecutive days. It was considered that middle-range smokers take 0.4 mg/kg of nicotine per day [36]. The concentration of 0.18 mg/kg diazinon is below no observed adverse effect level (NOAEL) in mouse [34]. The nicotine and diazinon were dissolved in 0.2 mL of corn oil (Cat#: C8276, Sigma-Aldrich) as a vehicle. After weaning, the female and male offspring were separated into groups of 3–5 animals. All the procedures were approved by the Institutional Animal Care and Use Committee of Chungbuk National University (CBNUA-1510-21-01).

### 4.5. Immunohistochemistry

Offspring mice were anesthetized using Avertin (2,2,2-tribromoethanol: T48402; Sigma-Aldrich with Ter-amyl alcohol: 240486; Sigma-Aldrich). The brains were harvested and briefly fixed in 4% paraformaldehyde (PFA) for 24 h. Then, the brains were transferred to phosphate-buffered 30% sucrose for 24 h for dehydration. The tissues were embedded with a cryopreservation compound (Cat#:3801481, Leica Biosystems, Richmond, IL, USA) and snap-frozen in liquid nitrogen. Tissues were sectioned coronally with cryotome (Cat#: CM1860, Leica Biosystems) with 60 μm of thicknesses. The tissues were washed with PBS to remove the cryopreservation compound. The antigen retrieval process was performed with 2N hydrochloric acid (HCl) at 36.5 °C for 30 min. Then, tissues were neutralized with 1M borate buffer at RT for 10 min. Nonspecific reactions were blocked by incubating the sections in 10% normal goat serum (Vector Laboratories, Burlingame, CA, USA) in PBS for 1 h at room temperature. Then, tissues were incubated with primary antibodies against TH (1:500, Cat#: P24529, Aves Labs, Tigard, OR, USA). For fluorescent labelling, Alexa Fluor488 goat anti-chicken (1:1000, Cat#: A11039, Invitrogen, Carlsbad, CA, USA) were used. Then, tissues were mounted on slides with Fluoro-Gel (Cat#:17985-11, Emsdiasum, Hatfield, PA, USA). All images were obtained using the Lionheart FX Automated Microscope (BioTek).

### 4.6. Measurement of the TH-Positive Cells on the Ventral Tegmental Area

For evaluating the three or four consecutive tissue slides obtained around −3 mm brain position were prepared and immunolabelled. All the tissues were simultaneously immunolabelled. The microscopic images of the brain tissues were converted into 8-bit grayscale using NIH Image J software. The images were adjusted to have an equal threshold level. Then, the number of cell bodies were counted as TH-positive dopaminergic neurons. The number of cells per mouse was determined by triplicate trials with consecutive tissues.

### 4.7. Experimental Behavior Analysis

For the assessment of animal behavior, offspring mice between the age of 4 weeks and 10 weeks were arbitrarily selected to perform the behavior tests as previously described [37,38]. All the behavior tests were undertaken by separating the sexes. The number of mice and details for all behavioral tests are described in Table 1.

### 4.8. Rotarod Test

Mice were placed on a rotarod apparatus (Panlab, Sydney, Australia) and the latency to fall off the rod was observed. The speed of the rotarod accelerated from 4 to 40 rpm over a term of 5 min. Mice were given three trials with at least 20 min of recovery time between trials. Mice received a two-phase training protocol for 10 days: training phase (3 consecutive days) and followed by test phase (every two days from day 4 to 10).

### 4.9. Open Field Test

For the open field test, mice were placed on a cube-shaped space 50 cm tall, 60 cm in width, and with a white-colored bottom. At first, each mouse was individually located toward the wall-side, then allowed to move freely for 5 min to estimate the locomotion. All the trials were recorded with a video camera and the recorded files were analyzed with the EthoVision XT14 software (Noldus, Leesburg, VA, USA). Trips, velocity, and suspending time in the center area were evaluated.

### 4.10. Marble Burying Test

A mouse was placed individually in the polycarbonate cage (25 × 40 cm width with 18 cm height) containing 5 cm-deep bedding (corn cob) for 10 min. Then, the mouse was removed from the cage and 20 glass marbles (15 mm diameter) were laid on the surface of the bedding equidistantly from each other in a 4 × 5 arrangement. The mouse was placed into a corner of the cage and allowed to bury marbles for 30 min. After the test, the number of buried marbles was counted. The marbles covered at least 50% by the bedding were considered buried.

### 4.11. Nest Building Test

Mice were placed in each individual cage and assigned nesting with cotton material (5 × 5 cm width, weight 2.5 g) already placed in the cage. Then, mice were allowed to complete 1 trial for 12 h undisturbed. After the mouse was carefully removed from the cage, each nest in the cages was photographed. Nest building ability was evaluated according to a 5-score rating scale. All mice were investigated simultaneously and the mean score of the total trials was considered.

### 4.12. Tail Suspension

Mice were suspended on the edge of a shelf, 50 cm above the surface of a table, using adhesive tape. The subject mouse allowed to move for 6 min. The duration of immobility was recorded in the last 5 min. The behavior was recorded by a video camera and analyzed using EthoVision XT14.

### 4.13. Morris Water Maze

The Morris water maze was performed in a circular swimming pool (a diameter of 90 cm with 40 cm height), filled with water at 25 °C, and skim milk was added to make the water translucent. The swimming pool was divided into four sections equally (I, II, III, IV). A platform was placed in the middle of a target quadrant III (12 cm from the pool’s edge and 1 cm below the water surface) with visual cues on the wall as spatial references. Mice received a two-phase training protocol for 9 consecutive days: cue training (4 days) followed by spatial training (5 days). Four trials were conducted per mouse per day, and the escape latency (time to find the platform beneath the water surface) in each trial was recorded. For each trial, the subject mouse was gently placed into the water facing toward the wall from one of three quadrants, varied by days of trial. Each mouse was given 60 s to find the platform. If any mouse failed to find the platform within 60 s, the mouse was gently guided to the platform by the investigator and allowed to rest on the platform for 30 s. The mean of escape latencies for four trials is represented for the learning result for each mouse. On day 10, the platform was removed from the pool, and the mice were left to search for the platform for 60 s. Videos were recorded and analyzed using EthoVision XT14. The time spent in the target platform area and the number of platform area crossings were observed to indicate memory ability.

### 4.14. Nobel Object Recognition Test

The subject mouse was exposed to two identical objects for 10 min. Then, one of the objects was replaced by a novel object, and the subject mouse was allowed to explore the novel object for another 10 min. The duration that mice spent interacting with the novel object and the previous object was observed and analyzed using EthoVision XT14.

### 4.15. Three-Chamber Sociability Test

The three-chambered social test was conducted as previously described. The apparatus for the three-chamber sociability test consisted of three Plexiglass-chambers. Each chamber has 40 cm width, 20 cm length, and 22 cm tall with small openings (10 cm × 5 cm) that allow mice to access each chamber. At first, the subject mouse was placed in the middle chamber and allowed to freely explore all three chambers with an empty circular wired cup (a diameter of 8 cm with 10 cm height with gride wires) in each side chamber for the 5 min habituation time. For sociability testing, a stranger mouse (stranger1, referred to as ‘S1’) was then introduced in a cylindrical plastic cage in one of the side chambers and an empty cylindrical plastic cage (empty, referred to as ‘E’) on the other side of the chamber. Then, the subject mouse was placed into the middle chamber and allowed to freely explore all three chambers for 10 min. For the social novelty test, the empty plastic cage was replaced with a wild-type stimulus mouse (stranger1, referred to as ‘S2’) and the subject mouse again explored all three chambers for 10 min. Time spent closely with the cages, travelling distance, and heat maps were measured using the EthoVision XT14. The preference index for each animal was calculated as:$$Preference\;index=\;\frac{\left(S1-E\right)}{\left(S1+E\right)}\;or\;as\;\frac{\left(S2-S1\right)}{\left(S2+S1\right)}$$
where ‘*E*’, ‘*S*1’, and ‘*S*2’ represent the time spent closely with empty cage, the stranger1, or 2, respectively.

### 4.16. Social Interaction

The mouse was introduced in the open field apparatus with the presence of a stranger mouse and allowed to explore freely for 10 min. The social interaction indexes, such as general sniffing, anogenital sniffing, following, mouthing, and fighting, were observed.

### 4.17. Data Analysis

All data were analyzed by applying the normality test followed by analysis of variance (ANOVA) followed by Turkey’s test for multiple comparisons. Litter was used as the unit of variance for treatment effects in ANOVA, with sex as a within litter factor. Statistical analysis was performed by using Graph Pad Prism (v.5.0; GraphPad Software, San Diego, CA, USA).

## Figures and Tables

**Figure 1 ijms-22-07742-f001:**
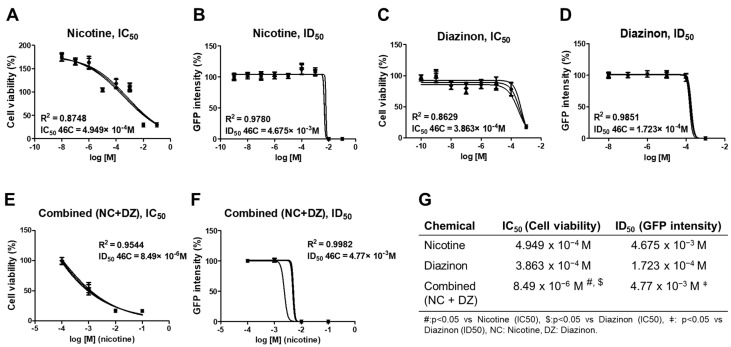
Synergistic adverse effect of combined exposure to nicotine and diazinon in vitro. (**A**–**F**), Sox1-GFP cell viability assay, IC_50_ value was measured with CCK-8 assay and ID_50_ value was detected by the intensity of green fluorescent protein (GFP); IC_50_, the half maximal inhibition concentration; ID_50_, the half inhibition concentration for neural differentiation; (**A**), IC_50_ value of nicotine; (**B**), ID_50_ value of nicotine; (**C**), IC_50_ value of diazinon; (**D**), ID_50_ value of diazinon; (**E**), IC_50_ value of the combined exposure to nicotine and diazinon; (**F**), ID_50_ value of the combined exposure to nicotine and diazinon; R^2^, coefficient of determination, the data were presented with standard variation; (**G**), the summery of the values and the statistical significance, # *p* = 0.0283; $ *p* = 0.0142, ǂ *p* = 0.047. Data are presented as the means ± standard deviation of three separate experiments. Statistical significance was determined by two-tailed *t*-test with the Welch correction test.

**Figure 2 ijms-22-07742-f002:**
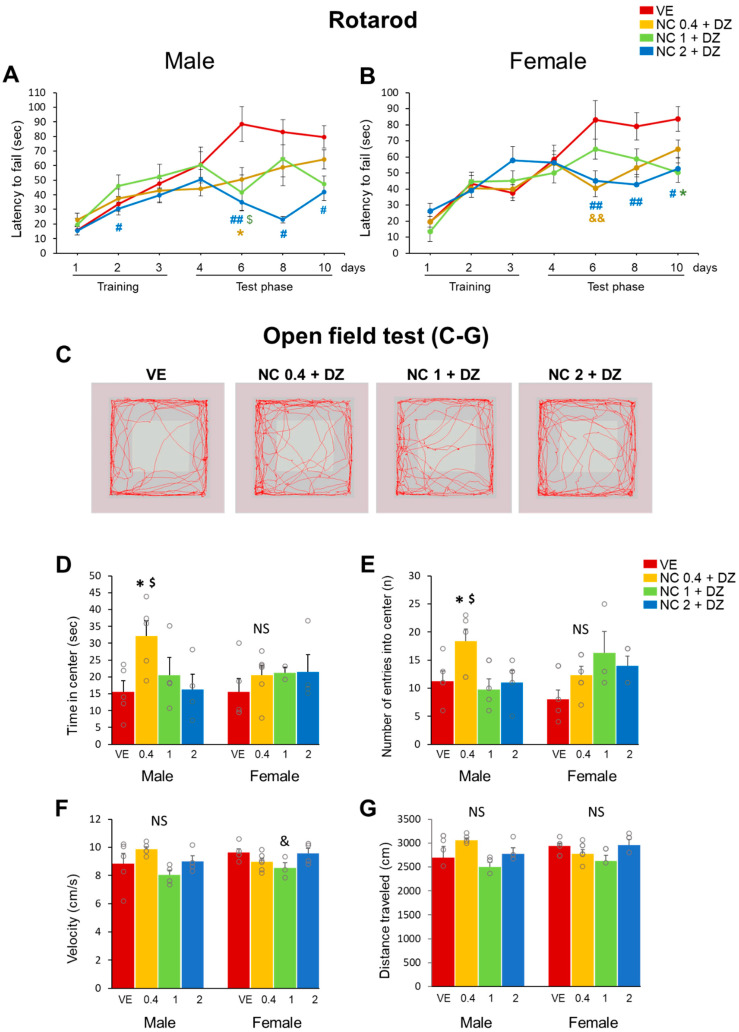
Combined exposure to nicotine and diazinon impaired locomotor activity in male and female mice. (**A**), rotarod test for male mice, the latency to fall during test phases decreased in male treated group compared to VE. NC0.4 + DZ (day 6: * *p* < 0.05) NC1 + DZ (day 6: $ *p* < 0.05) and NC2 + DZ-treated mice (day 2: # *p* < 0.05, day 6: ## *p* < 0.01, day 8: # *p* < 0.05, day 10: # *p* < 0.05); (**B**), rotarod test for female, similar to male mice, chemical-exposed group exhibited motor latency during test phase compared to VE, NC0.4 + DZ (day 6: && *p* < 0.01), NC1 + DZ (day 10: * *p* < 0.05), NC2 + DZ (day 6: ## *p* < 0.01, day 8: ## *p* < 0.01, day10: # *p* < 0.05); (**C**), tracing schematics of mouse travel in the open field test. (**D**–**G**) are quantification of open field result; **D**, time in center, male NC0.4 + DZ group spent more time in center of the open field (* *p* < 0.05 vs. VE, $ *p* < 0.05 vs. NC2 + DZ, *F*_3,20_ = 5.529, *p* = 0.0089), while female group did not show any significance; (**E**), number of entries into center; male mice in NC0.4 + DZ treated group showed frequent access to the center than VE (* *p* < 0.05 vs. VE, $ *p* < 0.05 vs. NC2_DZ, *F*_3,20_ = 6.173 *p* = 0.054), while that of female group are not significant; (**F**), velocity, not significant, NC1 + DZ vs. NC2 + DZ in female mice (& *p* < 0.05, *F*_3,20_ = 3.636 * *p* = 0.0328); (**G**), Distance traveled, No significant differences in both male and female mice. VE, vehicle-treated group; NC0.4 + DZ, 0.4 mg/kg nicotine and 0.18 mg/kg diazinon-treated group; NC1 + DZ, 1 mg/kg nicotine and 0.18 mg/kg diazinon-treated group; NC2 + DZ, 2 mg/kg nicotine and 0.18 mg/kg diazinon-treated group. NS: no significance, Data are presented as the means ± standard error of the mean. Statistical significance was determined by one-way ANOVA with the Tukey’s correction test.

**Figure 3 ijms-22-07742-f003:**
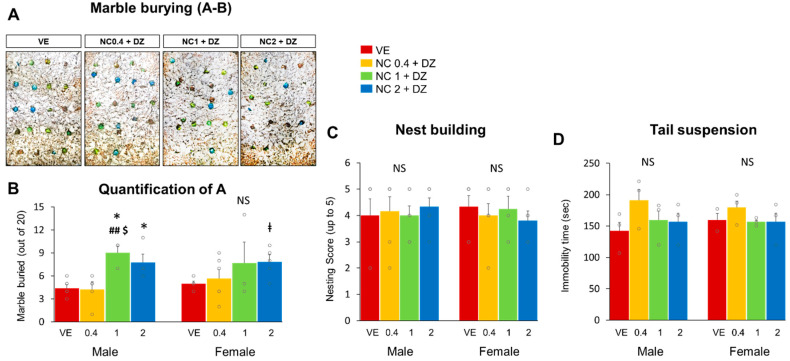
Co-exposure to nicotine and diazinon increased compulsive-like behavior in mice. (**A**), representative images of marble-burying test; (**B**), quantification of A, in the marble-burying test, mice in NC 0.1 + DZ and NC2 + DZ treated group buried significantly increased number of marbles in both males and females (male mice: * *p* < 0.05 vs. VE, ## *p* < 0.05 vs. NC0.4 + DZ, $ *p* < 0.05 vs. NC2 + DZ, *F*_3,20_ = 7.322, *p* = 0.004, female mice: ǂ *p* < 0.05 vs. VE, *F*_3,20_ = 4.110, *p* = 0.012); (**C**), nest-building test result, no significance; (**D**), tail suspension test result, no significance; VE, vehicle-treated group; NC0.4 + DZ, 0.4 mg/kg nicotine and 0.18 mg/kg diazinon-treated group; NC1 + DZ, 1 mg/kg nicotine and 0.18 mg/kg diazinon-treated group; NC2 + DZ, 2 mg/kg nicotine and 0.18 mg/kg diazinon-treated group; NS, no significance. Data are presented as the means ± standard error of the mean. Statistical significance was determined by one-way ANOVA with the Tukey’s correction test.

**Figure 4 ijms-22-07742-f004:**
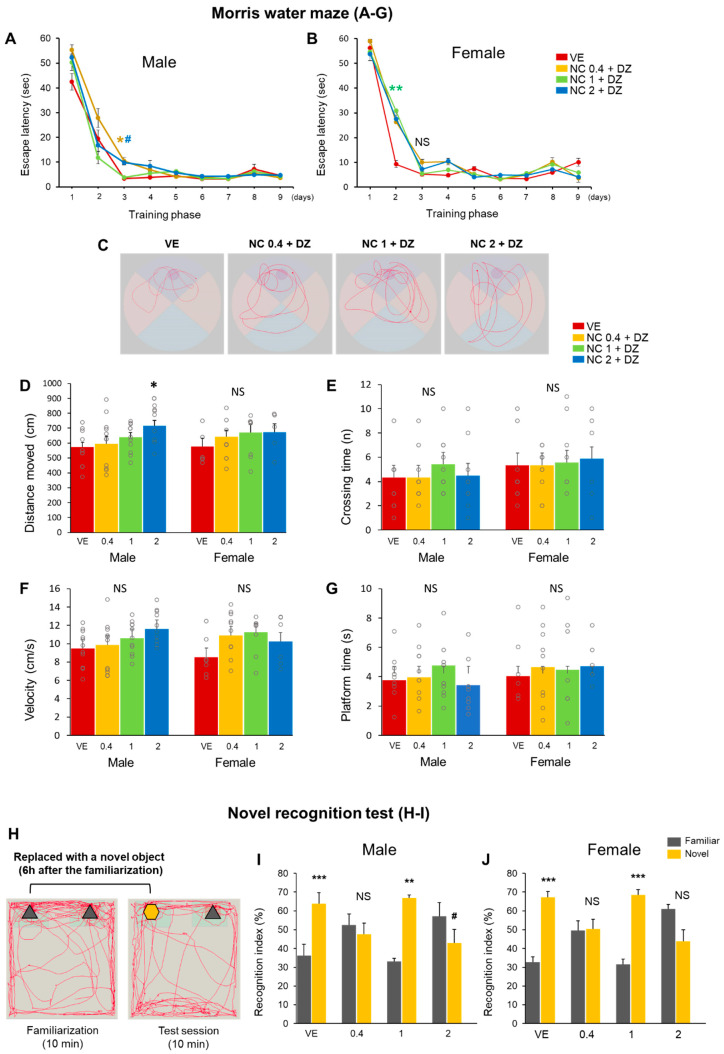
Combined exposure to nicotine and diazinon disrupted spatial reference learning and novel recognition behaviors. (**A**), Morris water maze test for male offspring, NC0.4 + DZ and NC2 + DZ-treated group showed relatively greater delay for escape time during the initial phase of leaning compared to mice from the control group, (day 3: * *p* < 0.05 vs. NC0.1 + DZ vs. VE; # *p* < 0.05, NC2 + DZ vs. VE, *F*_3,64_ = 6.197, *p* = 0.004); (**B**), Morris water maze test for female offspring, on day 2, NC1 + DZ injected mice exhibited a reduced spatial memory learning than that of VE (** *p* < 0.01 vs. VE, *F*_3,64_ = 5.101, *p* = 0.008). (**C**), Representative swimming path, (**D**–**G**), Quantification of B, Only in the male offspring, NC2 + DZ treated mice swam a longer distance during the test compared to that of VE (* *p* < 0.05 vs. VE, *F*_3,48_ = 2.72, *p* = 0.050); (**H**), schematic images of the novel object recognition test; (**I**), recognition index for male offspring, NC0.4 + DZ and NC2 + DZ-treated mice showed abnormal recognition behavior in novel object test (** *p* < 0.01, *** *p* < 0.001 increased recognition index for familiar object vs. novel object; NS, no significance, # *p* < 0.05 decreased recognition index familiar object vs. novel object); (**J**), recognition index for female offspring. NC0.4 + DZ and NC2 + DZ exposure mice exhibited abnormal recognition behavior (*** *p* < 0.001 increased recognition index for familiar object vs. novel object, NS: no significance); VE, vehicle-treated group; NC0.4 + DZ, 0.4 mg/kg nicotine and 0.18 mg/kg diazinon-treated group; NC1 + DZ, 1 mg/kg nicotine and 0.18 mg/kg diazinon-treated group; NC2 + DZ, 2 mg/kg nicotine and 0.18 mg/kg diazinon-treated group. Data are presented as the means ± standard error of the mean. Statistical significance was determined by two tailed T-test and one-way ANOVA with the Tukey’s correction test.

**Figure 5 ijms-22-07742-f005:**
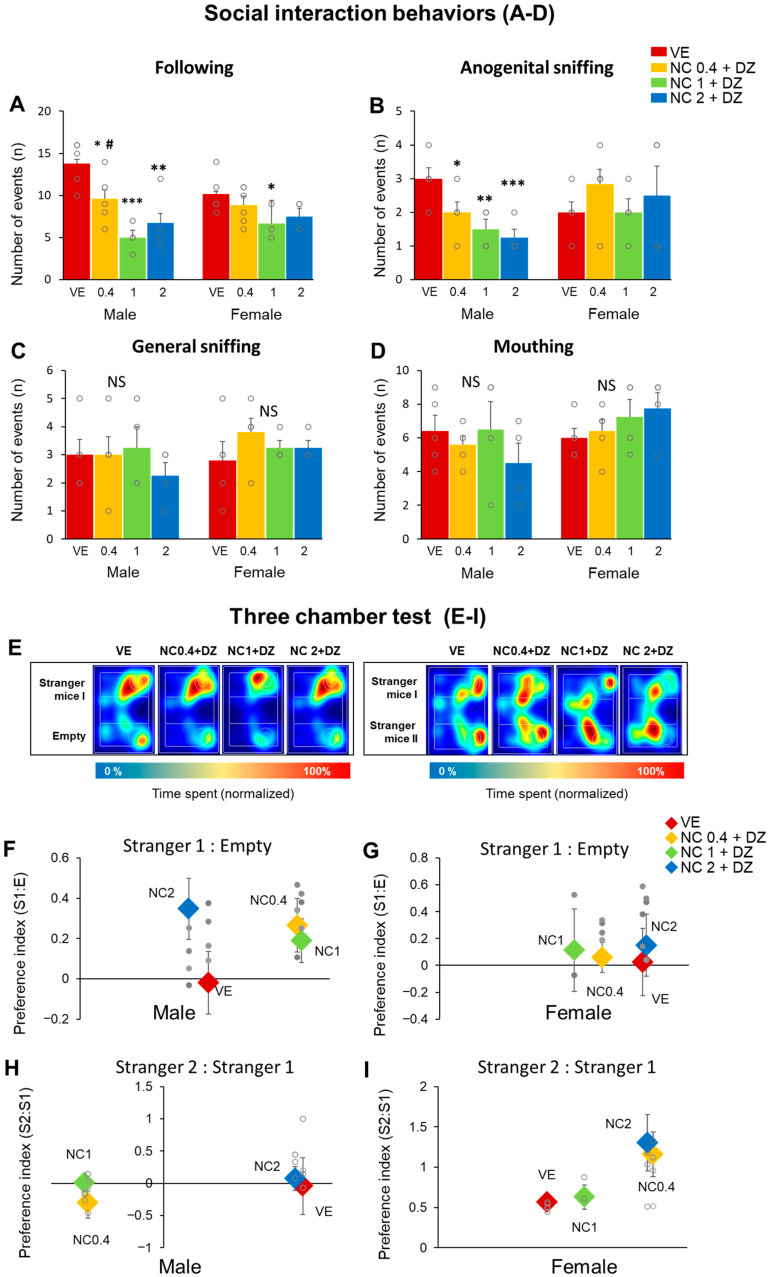
Combined administration of nicotine and diazinon influenced mouse social interaction behaviors, but not influenced on three-chamber social interaction. (**A**–**D**), social interaction test results. (**A**), following behavior, male mice in all three chemical-treated group showed lower following behavior (* *p* < 0.05, ** *p* < 0.01, *** *p* < 0.001 vs. VE, # *p* < 0.05 NC0.4 + DZ vs. NC1 + DZ, *F*_3,20_ = 13.07, *p* < 0.0001), only NC1 + DZ treatment group exhibited significant reduction in following behavior (* *p* < 0.05 vs. VE, *F*_3,20_ = 3.465, *p* = 0.038); (**B**), anogenital sniffing, nicotine and diazinon treatment resulted in decreased sniffing behavior in all three experimental group (* *p* < 0.05, ** *p* < 0.01, *** *p* < 0.001 vs. VE, *F*_3,20_ = 10.45, *p* = 0.0003); while none of the female offspring show any changes in the test result; (**C**), general sniffing, there was no significant difference compared to VE in each group; (**D**), mouthing behavior, no statistical significance was detected; (**E**), Representative heat map images from three-chamber sociability test, (**F**–**I**), Preference indexes were calculated from three chamber behavior. In male offspring, NC0.4 + DZ and NC2 + DZ exposure group indicated the tendency to not prefer to stranger mice, but statistical significances were not detected. VE, vehicle-treated group; NC0.4 + DZ, 0.4 mg/kg nicotine and 0.18 mg/kg diazinon-treated group; NC1 + DZ, 1 mg/kg nicotine and 0.18 mg/kg diazinon-treated group; NC2 + DZ, 2 mg/kg nicotine and 0.18 mg/kg diazinon-treated group. Data are presented as the means ± standard error of the mean. Statistical significance was determined by one-way ANOVA with the Tukey’s correction test.

**Figure 6 ijms-22-07742-f006:**
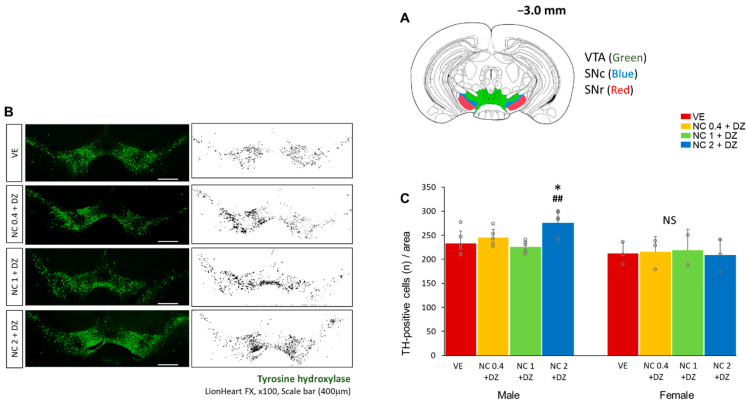
Co-exposure to nicotine and diazinon changed the number of dopaminergic neurons in male mice brain. Brain tissues from male mice offspring (n = 4 for male, n = 3 for female) were immunolabelled with anti-tyrosine hydroxylase (TH) antibody. (**A**), schematic images of the selected brain region, ventral tegmental area (VTA: green colored), substantia nigra pars compacta (SNc: blue colored), substantia nigra pars reticulata (SNr: red colored); (**B**), localization of Tyrosine hydroxylase (TH)-positive neurons in coronal brain section; images were converted to 8-bit grey-scale via NIH Image, scale bar, 400 μm; (**C**), Quantification of B, the amount of TH-positive cells increased in the NC2 + DZ treated male group compared to vehicle and NC1 + DZ exposure group (* *p* < 0.05 vs. VE, ## *p* < 0.01 vs. NC1 + DZ, *F*_3,24_ = 5.646*, p* = 0.0057). No significance was detected among female groups. VE, vehicle-treated group; NC0.4 + DZ, 0.4 mg/kg nicotine and 0.18 mg/kg diazinon-treated group; NC1 + DZ, 1 mg/kg nicotine and 0.18 mg/kg diazinon-treated group; NC2 + DZ, 2 mg/kg nicotine and 0.18 mg/kg diazinon-treated group. Female staining data were not presented due to no statistical significance (Data are not shown). Data are presented as the means ± standard deviation. Statistical significance was determined by one-way ANOVA with the Tukey’s correction test.

**Table 1 ijms-22-07742-t001:** Animal data for behavior tests.

Test	Measurement	Number of Animals per Group (Sex)	Age (Old)
Rotarod	Latency to fall (s)	*n* = 10 (5 male, 5 female) for VE, *n* = 10 (5 male, 5 female) for NC0.4 + DZ, *n* = 7 (4 male, 3 female) for NC1 + DZ, *n* = 8 (4 male, 3 female) for NC2 + DZ	6~8 weeks
Open field	Time in center (s)	*n* = 10 (5 male, 5 female) for VE, *n* = 10 (5 male, 5 female) for NC0.4 + DZ, *n* = 7 (4 male, 3 female) for NC1 + DZ, *n* = 8 (4 male, 3 female) for NC2 + DZ	4 weeks
	Frequency (*n*)		
	Velocity (cm/s)		
	Distance traveled (cm)		
Marble burying	Number of marbles buried (*n*)	*n* = 10 (5 male, 5 female) for VE, *n* = 10 (5 male, 5 female) for NC0.4 + DZ, *n* = 7 (4 male, 3 female) for NC1 + DZ, *n* = 8 (4 male, 3 female) for NC2 + DZ	9 weeks
Nest building	Nesting score	*n* = 7 for each group (4 male, 3 female)	4 weeks
Tail suspension	Immobility time (s)	*n* = 7 for each group (4 male, 3 female)	9 weeks
Morris water maze	Escape latency (s)	*n* = 8 (4 male, 4 female) for VE, *n* = 8 (4 male, 4 female) for NC0.4 + DZ, *n* = 7 (4 male, 3 female) for NC1 + DZ, *n* = 8 (4 male, 3 female) for NC2 + DZ	9~10 weeks
Noble object recognition	Recognition index (%)	*n* = 10 (5 male, 5 female) for VE, *n* = 10 (5 male, 5 female) for NC0.4 + DZ, *n* = 7 (4 male, 3 female) for NC1 + DZ, *n* = 8 (4 male, 3 female) for NC2 + DZ	5 weeks
Three-chamber sociability	Time spent in chamber (s)	*n* = 10 (5 male, 5 female) for VE, *n* = 10 (5 male, 5 female) for NC0.4 + DZ, *n* = 7 (4 male, 3 female) for NC1 + DZ, *n* = 8 (4 male, 3 female) for NC2 + DZ	5 weeks
Social interaction	Following (*n*) Genital sniffing(*n*) Anogenital sniffing (*n*) Mouthing	*n* = 10 (5 male, 5 female) for VE, *n* = 10 (5 male, 5 female) for NC0.4 + DZ, *n* = 7 (4 male, 3 female) for NC1 + DZ, *n* = 8 (4 male, 3 female) for NC2 + DZ	9 weeks

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
