# Peer review of "Combined Exposure to Diazinon and Nicotine Exerts a Synergistic Adverse Effect In Vitro and Disrupts Brain Development and Behaviors In Vivo"

_ijms, 2021, doi:10.3390/ijms22147742_

Round 1
Reviewer 1 Report
The manuscript "Combined Exposure to Diazinon and Nicotine Exerts a Synergistic Adverse Effect in vitro and Disrupts Brain Development and Behaviors in vivo." is an interesting and well-written study. Following modification is required to improve the manuscript quality.
Comments
Is figure 2C experiment was conducted with only NC.
The open-field test result shows a significant difference between males and females but not explained the cause in the discussion part.
The study well describes that the combined exposure to Diazinon and Nicotine exerts synergistic adverse effects, but it does not compare to Diazinon and Nicotine individually. I would recommend including Diazinon alone and Nicotine alone exposure results to compare it with the combined exposure for better justification.
Include female staining results in figure 6 even if it is not significant.
Author Response
Reviewer 1.
The manuscript "Combined Exposure to Diazinon and Nicotine Exerts a Synergistic Adverse Effect in vitro and Disrupts Brain Development and Behaviors in vivo." is an interesting and well-written study. Following modification is required to improve the manuscript quality.
We would like to thank you for your constructive suggestion on this manuscript. The manuscript has been revised after deeply considering your review report. With your comments, we tried to make improve our manuscript to elaborate. We also added more descriptions of methods and more references. All the modifications were marked with ‘track changes.' Please find and review our revised manuscript. Thank you.
Comments
Is figure 2C experiment was conducted with only NC.
The 2C was conducted with both nicotine and diazinon. To avoid confusion, we revised the legends to show both NC and DZ.
The open-field test result shows a significant difference between males and females but not explained the cause in the discussion part.
As you recommended, the open-field data had been not fully explained. We added a reference to elaborate the behavior result of perinatal nicotine exposure (Zhou et al, 2021: doi:10.389/fphar.2021.661304). We revised our manuscript to explain our open-field result. Please find the revised sentences as below:
“Zhou (2021) revealed that maternal nicotine exposure (via drinking water; 200 g/mL) increased in moving distance in an open field test for offspring mice (Zhou et al., 2021).” (Line 331)
The study well describes that the combined exposure to Diazinon and Nicotine exerts synergistic adverse effects, but it does not compare to Diazinon and Nicotine individually. I would recommend including Diazinon alone and Nicotine alone exposure results to compare it with the combined exposure for better justification.
We fully agree with your suggestion that single chemical exposure data is required to demonstrate a synergistic effect in vivo. With all due respect, it is difficult to re-do the experiment due to a lack of resources. Albeit we intended to support our hypothesis by presenting the single chemical exposure data in vitro, it might be insufficient. For now, this was the best.
However, we did not want to aggrandize or mislead readers. To avoid any confusion, we revised our sentences regarding the synergism:
“For example, a combination of benzopyrene and lead caused spatial learning and memory impairments by amplifying oxidative stress levels (Youbin et al., 2013). (Line 406)”
“The present study indicates there is synergism in the action of nicotine and diazinon both in vitro, and perinatal exposure to both nicotine and diazinon resulted in abnormal mice behaviors and alteration in TH-positive neurons in the brain. (Line: 413)“
Include female staining results in figure 6 even if it is not significant.
As you recommended, we added the female staining data. No statistical significance was detected among female mice groups; however, there was a sex difference between NC2+DZ treated male and female mice. Please find the revised figure 6.

Reviewer 2 Report
The present study evaluates the in vitro and in vivo effects of combinations of nicotine and diazinon. The in vitro studies are straightforward and support the claim that there are interactive effects of the two drugs on cell viability.
A battery of behavioral tests were conducted in the offspring of mice that were treated with a combination of diazinon and nicotine at three dose levels for four days during pregnancy. Data interpretation for this set of studies is problematic. The authors cannot claim (line 386) that there are synergistic effects of the two drugs since there were no nicotine alone control groups. Furthermore, since sex was not included as a factor in ANOVA analyses, the authors cannot state that there were significant sex differences. The authors also make the misleading claim that low doses of nicotine were used (line 376), which is not true.
Author Response
Reviewer 2
The present study evaluates the in vitro and in vivo effects of combinations of nicotine and diazinon. The in vitro studies are straightforward and support the claim that there are interactive effects of the two drugs on cell viability.
We would like to appreciate your constructive advice on our manuscript. The manuscript has been revised after deeply considering your review report. With your comments, we tried to make improve our manuscript to elaborate. We also added more descriptions of methods. All the modifications were marked with ‘track-changes.’ Please find and review our revised manuscript. Thank you.
A battery of behavioral tests were conducted in the offspring of mice that were treated with a combination of diazinon and nicotine at three dose levels for four days during pregnancy.
Previously, we wrote the introduction briefly, considering that nicotine or diazinon have been documented individually. However, it might be insufficient to provide critical information to readers. As you advised, the dose of nicotine and the concept of drug synergism were unclear, which may mislead the readers. Thus, we revised our manuscript to elaborate on our experimental design and research aim. Additionally, we added some descriptions regarding the methods. Please find our revised manuscript.
Data interpretation for this set of studies is problematic. The authors cannot claim (line 386) that there are synergistic effects of the two drugs since there were no nicotine alone control groups.
We fully agree with your suggestion. Admittedly, albeit we could demonstrate a synergistic effect of the two chemicals in vitro, nicotine and diazinon alone exposure data should be required to substantiate the in vivo results. We revised the sentences not to mislead readers. The revised sentences are as below:
“For example, a combination of benzopyrene and lead caused spatial learning and memory impairments by amplifying oxidative stress levels (Youbin et al., 2013). (Line 406)”
“The present study indicates there is synergism in the action of nicotine and diazinon both in vitro, and perinatal exposure to both nicotine and diazinon resulted in abnormal mice behaviors and alteration in TH-positive neurons in the brain. (Line: 413)“
Furthermore, since sex was not included as a factor in ANOVA analyses, the authors cannot state that there were significant sex differences.
As you suggested, sex was not included as a factor in ANOVA analyses. However, we observed that the chemical exerted an effect in a sex-dependent manner. For example, in an open field test, even though there was no statistical difference in variances among the male and female groups, only male mice were influenced by the chemical exposure, while females did not exhibit an alteration. In this regard, with all due respect, a comparison of the means between males and females can elaborate on the different responses to the chemicals between the two sexes.
However, we fully agree that was an inappropriate method to explain the statistical differences between the two sexes. To avoid any confusion, we revised the sentences that there was a significant sex difference as below:
Male mice were more susceptible to perinatal nicotine and diazinon exposure, while fe-male offspring did not exhibit a significant alteration. (Deleted, Line 237)
“Male mice exhibited an increase in the number of events in NC2 + DZ treated group” (Line 262)
“Female showed a decrease in the mean of the number of cells in NC2 + DZ treated group (ǂp=0.02).” (Line 297)
“We also confirmed that nicotine and diazinon altered the offspring behavior involving motor learning, anxiety-like behavior, and social interaction patterns in a sex-dependent manner.” (Line 344)
“Data exhibiting any sex-dependent differences in the means between males and females were also marked up in Table 1.” (Line 508)
“Mean comparison by sex” (Table 1)
The authors also make the misleading claim that low doses of nicotine were used (line 376), which is not true.
0.4mg/kg of nicotine has been commonly used in previous nicotine studies such as Jennifer et al (doi: 10.1016/j.pbb.2011.03.009), Ge Li-Sha et al (doi:10.1016/j/lif.2016.02.003). Compared to those previous works, this study used a relatively short period, thus we insisted that the dose was relatively low. However, as you mentioned, it is not true that the dose of nicotine is low per se, which may mislead readers. To make it clear, we revised the expression that the dose of nicotine was low. Additionally, in our discussion part, we already discussed other nicotine-related studies and described the dose of nicotine used for the studies. We intended to help readers to compare the results with our data. The revised sentence is as below:
“Our combined administration of nicotine and diazinon reproduced similar phenotypes to those observed in the above studies, even though our treatment involved low-doses and brief exposure periods for nicotine with a NOAEL dose of diazinon.” (Line 352)
Again, we appreciate your constructive suggestions on this manuscript.

Round 2
Reviewer 1 Report
The manuscript "Combined Exposure to Diazinon and Nicotine Exerts a Synergistic Adverse Effect in vitro and Disrupts Brain Development and Behaviors in vivo" now looks very interesting. I appreciate the author's effort to revise the manuscript.
Minor Correction
The authors did not include single chemical exposure data in the revised manuscript due to a lack of resources. Therefore, I recommend authors include this as a limitation of the study in the manuscript.
Author Response
The manuscript "Combined Exposure to Diazinon and Nicotine Exerts a Synergistic Adverse Effect in vitro and Disrupts Brain Development and Behaviors in vivo" now looks very interesting. I appreciate the author's effort to revise the manuscript.
We would like to thank you for your appreciation and advice on the manuscript. Due to your pinpoint suggestions, we could ameliorate this manuscript with more clear data presentation and explanations. We highly appreciate your effort to review this.
Minor Correction
The authors did not include single chemical exposure data in the revised manuscript due to a lack of resources. Therefore, I recommend authors include this as a limitation of the study in the manuscript.
The manuscript has been revised to mention the limitation of this study, in the discussion part. The revised sentence is as below:
“However, there is a limitation that nicotine or diazinon single-exposure data was not included in the study; the seriousness of exposure to respective chemicals and the mechanism through which they act was not elucidated in the present study.” (Line 412 on viewing track changes)
Reviewer 2 Report
The authors have made substantial changes in response to the prior reviews. However, they continue to draw comparisons between sexes without demonstrating that sex is a significant factor in an overall ANOVA. This must be done before such comparisons can be made.
Author Response
The authors have made substantial changes in response to the prior reviews. However, they continue to draw comparisons between sexes without demonstrating that sex is a significant factor in an overall ANOVA. This must be done before such comparisons can be made.
As you recommended, we revisited our data and re-conducted statistical analyses including the sex variable. We found out that no significance was detected between the two sexes by ANOVA. Previously, we drew a conclusion that the chemicals exerted effects in a sex-dependent manner; however, we admit that the nature of the male and female data was not statistically different. Thus, we revised the manuscript to focus on the result per se, without highlighting sex differences. In the discussion part, we mentioned that no significance was detected between the two sexes. Instead, we referred to previous studies that indicate a correlation between sex and behavioral impairments. In addition to this, we modified the table1 and methods. Please find and re-consider our manuscript.
“In addition, the chemical-treated mouse offspring showed abnormalities in motor learning, compulsive-like behaviors, spatial learning, and social interaction patterns.” (Line 23)
“Statistical significance of the data was not detected between the two sexes.” (Line 230 on viewing track changes)
“Furthermore, albeit we analyzed the data by splitting males and females, we could not demonstrate statistical difference by the sex variable. However, previous reports have shown a correlation between offspring sex and impairment in offspring (Alkam et al., 2013; Cross et al., 2017; Rebuli & Patisaul, 2016; Vatanparast et al., 2013).” (Line 336)
“At last, prenatal diazinon exposure altered the passive avoidance performance and neuronal nitric oxide (nNOS) synthase neurons in basolateral complex of amygdala in female rat, while the males were not affected (Vatanparast et al., 2013).” (Line 354)
“Data exhibiting differences in the means between males and females were also marked up in Table1.” (deleted; line 504)
“Litter was used as the unit of variance for treatment effect in ANOVA, with sex as a within litter factor.” (Line590)
In addition to the changes above, we revised some parts of the manuscript. Please find the manuscript on viewing track changes. Thank you very much.
Round 3
Reviewer 2 Report
The authors have changed the text extensively to eliminate discussion of sex differences. However, the figures show data still separated by sex with sex-dependent statistical analyses. This is not appropriate if there are no sex effects detected in the overall ANOVA and is somewhat confusing since mention of sex differences has been eliminated from the text. The paper would benefit from collapsing sex in the figures.
This manuscript is a resubmission of an earlier submission. The following is a list of the peer review reports and author responses from that submission.
Round 1
Reviewer 1 Report
The paper"Combined Exposure to Diazinon and Nicotine Exerts a Synergistic Adverse Effect in vitro and Disrupts Brain Development and Behaviors in vivo" is a fascinating study and well-written manuscript. The current investigation suggests that the combined exposure to diazinon and nicotine disrupts brain development and behaviors.
Minor comments
In figure 2, mark statistically significant changes.
Provide the version of statistical software in methods.
Author Response
We would like to thank you for the constructive suggestions on how to improve the quality of this manuscript. As you suggested, we marked statistical significance in figure2. Also, we clarified the version of the statistical software in methods.
“Statistical analysis was performed by using Graph Pad Prism (v.5.0; GraphPad Software San Diego, CA, USA).” (Line: 493)
Again, we thank you for your appreciation.

Reviewer 2 Report
They examined the effect of combined exposure to Diazinon and Nicotine on brain development and behavior. However, there are two major flaws in their experimental design. Thus, restructuring the experimental design is absolutely required.
- There are clear sex differences on the effect of exposure to nicotine or diazinon. Thus, combining the data of both sexes is not acceptable. Data should be shown separately.
- Although it is very important to examine the effect of combined exposure of environmental chemicals in vivo, such effect should be compared with the effect of single chemical exposure. This manuscript does not examine the effect of single chemical exposure in vivo. Such data should be added.
Author Response
We would like to thank you for the constructive suggestions on how to improve the quality of this manuscript.
First, we added supplementary figures that can show sex differences. Also, we revised our manuscript to show the reason we combined the data of both sexes. We combined the data of both sexes after we obtained the data because we could not find significant sex differences between obtained data (t-test). As you suggested, nicotine and endocrine-disrupting chemicals have been known to influence mice development in a sex-dependent manner. To make it clear, we revised our manuscript to display the sex differences.
“Since data obtained from male and female mice did not exhibit significant differences, the results were integrated into the figures. Figures for sex-separated data are presented as supplementary figures.” (Line 408; please find the attached PDF)
Second, we agree with your suggestion that data from a single chemical exposure could make our result more reliable. We tried to explain our result by comparing it to previous references with a single-chemical study. Behavioral test results for nicotine-single exposure have been generally performed and reported by previous researchers. In discussion, we compared our present data to behavior test results from previous reports. Albeit those studies are not able to directly compare to ours, we intended the readers to find the tendency of the results of a single-chemical exposure. For example, nest building, marble burying, social preference, and novel recognition tests were compared to Tursun et al (doi:10.1016/j.bbr.2012.10.058). Morris-water maze and rotarod test were compared to single-chemical exposure result of Jiayue et al (doi:10.1093/ntr/ntu178) and Angela et al (doi: 10.1016/j.bbr.2019.04.007) respectively. Additionally, we added one more reference (Jeremy et al; doi: 10.1152/ajpregu.00156.2017) to support our social interaction behavior data.
“In addition, Li (2015) reported that nicotine (via drinking water; 200 mg/mL, ab libitum) decreased social interaction behaviors in mice.” (Line 269)
Again, we thank you for your appreciation.

Round 2
Reviewer 2 Report
Very little change has been made from previous manuscript. Even if there is no difference between two sexes, it is not acceptable to combine data of two sexes. In fact, however, by looking at supplementary figures, there are several differences between male and female data. If there is no statistical significances among groups, they can simply increase the number of animals. Furthermore, experimental condition of previous studies are not fully consistent from current studies. Thus, it is not appropriate only to compare their studies with previous ones.